**Data Availability Statement:** All relevant data are within the paper and Supporting Information files.

**Funding:** The authors received no specific funding for this work.

# The burden of medical contraindications to corneal donation: Time for review

**Oliver Dorado-Cortez**[1,2], **Sylvain Poinard**[1,2], **Magali Epinat**[3], **Fanny Collange**[4], **Sandrine Ninotta**[1,5], **Paul Goin**[1,2], **Jean Luc Perrot**[6], **Philippe Gain**[1,2], **Graeme Pollock**[7,8], **Gilles Thuret**[1,2]*

1 Laboratory Biology, Engineering and Imaging for Ophthalmology, Health Innovation Campus, Faculty of Medicine, University Jean Monnet, Saint-Etienne, France, 2 Ophthalmology Department, University Hospital, Saint-Etienne, France, 3 Hospital Coordination for Organs and Tissues Retrieval, University Hospital, Saint-Etienne, France, 4 Clinical Epidemiology, Center for Clinical Investigation-Clinical Trials (CIC EC) 1408, Saint-Etienne, France, 5 Eye Bank, Auvergne Rhône Alpes French Blood Center, Saint-Etienne, France, 6 Dermatology Department, University Hospital, Saint-Etienne, France, 7 Centre for Eye Research Australia, Royal Victorian Eye and Ear Hospital, Melbourne, Australia, 8 Department of Ophthalmology, Department of Surgery, University of Melbourne, Melbourne, Australia

* gilles.thuret@univ-st-etienne.fr

## Abstract

Corneal graft (keratoplasty) is the most common allograft in the world, but the imbalance between the number of donors and the number of patients waiting for transplants is abysmal on a global scale and varies enormously from one country to another. The risk of transmission of systemic diseases from donor to recipient is demonstrably low. In over 50 years and an estimated 2.5 million transplants, only 8 cases of rabies, 2 cases of hepatitis B and 2 cases of Creutzfeldt-Jakob disease (CJD) have been documented. Conversely, other cases of rabies, HIV, hep C, hep B and CJD have not been transmitted via keratoplasty. The list of medical contraindications (CI) to corneal donation also includes diseases for which no actual, only theoretical, risk has been identified, in particular, neurodegenerative diseases, hematological malignancies, melanomas, tumors of the central nervous system, neoplastic meningitis and lymphangitic carcinomatosis. Their contribution to the reduction in the potential donor pool has not previously been investigated. We analyzed 45 months of exhaustive data from the hospital coordination for organ and tissue procurement at St-Etienne University Hospital (01/01/2020 to 06/09/2023). Out of the 2349 consecutive potential donors' files analyzed by the coordination team,1346 (57%) had an CI to donation. The identification of a neurodegenerative disease was the most frequent, accounting for 16% of the files examined and 29% of CIs. Of these, 75% were related to cognitive disorders. The 5 diseases or families of diseases for which there is only theoretical risk of transmission equated to a loss of 712 potential donors, corresponding to 30% of the files examined and 53% of all CIs. Of the 1003 deceased without CI, 738 families (74%) were contacted. No objection to donation was received in 52% of cases, enabling 385 procurements to be carried out. Removing these 5 CIs would have increased the number of donors by 71% (658 instead of 385). The potential pool of corneal donors is significantly restricted by a group of CIs introduced decades ago in response to a theoretical transmission of disease. A substantive amount of evidence now suggests that many CIs now need to be reviewed, modified or discarded

**Competing interests:** The authors have declared that no competing interests exist.

altogether. This approach will result in a highly significant worldwide increase in the availability of corneas for transplant and have an immediate and major impact in reducing corneal blindness across the globe. We propose that this reduction in CIs be accompanied by a prospective evaluation process, by allocating the corneas of these donors to patients aged 75 years and over, and by monitoring them for a minimum of 5 years.

## Introduction

Corneal transplantation, or keratoplasty, is one of the oldest and most frequent forms of human-to-human transplantation, with an estimated 200,000 transplants per year worldwide [1]. It restores corneal transparency and/or architecture lost through various pathological processes and aims to restore or improve vision. The benefits of successful corneal transplantation in the quality of life, social activity, general well-being and productivity of an individual are profound as are the benefits to a community as a whole. The need for corneal transplants is increasing worldwide due to better access to treatment, the ageing of populations, and advances in surgical techniques, making transplants safer and more effective. As a result, the benefits of corneal transplantation are being offered earlier and to a greater number of patients who are also experiencing improved outcomes.

Unfortunately, the rate of corneal donation worldwide remains insufficient to meet the need. In 2012–13, we established that the imbalance was abysmal on a global scale, with 1 available graft for every 70 patients on the waiting list [1] As the number of people that could benefit from corneal transplantation is increasing, the gap between availability and need continues to widen. While the potential corneal donor pool is relatively high compared to other forms of deceased donation (ischemic times to donation is generous, there is no upper age limit, while organization and infrastructure required for retrieval are not complex), there are many societal, spiritual, organizational and medical limitations to donation. However, one barrier to improving donation rates that has not been fully investigated is the limitation placed on the potential donor pool by certain medical contraindications (CI) to donation, and the justification of these contraindications when considering an evidence-based risk assessment versus the benefit of transplantation.

Iatrogenic transmission of serious or fatal systemic disease from corneal transplantation is an extremely rare event. A review of the world literature reveals only 13 reports of disease transmission since 1939 (excepting bacterial and fungal contamination and transmission), while it is conservatively estimated that between 2 and 2.5 million corneal transplants have been performed in this time [2] Eight of the 13 cases involved the transmission of rabies (from six donors) [3] There have been two cases of Hepatitis B transmission (from 2 donors) [4] and two cases of tumor transmission (both derived from donor intraocular carcinomas, a retinoblastoma and a choroidal metastasis [5, 6]). While ten suspected cases of CJD transmission have been reported since 1974 [2] in only one case was CJD confirmed in both the donor and recipient [7] While some have classified this single case as "definite transmission" [2], the difficulties in establishing definite cause and effect in CJD transmission means it remains, as the original author suggested, a "possible transmission" [7]

In contrast, cases of absence of transmission from infected donors have also been reported for rabies [8] HIV [9–11] HCV [12, 13] HBV [14], and CJD [15] Therefore, from both the epidemiological evidence and the anatomy and physiology of the cornea uniquely reducing possible vectors of transmission, the risks of transmission of disease is both rare (as in the case of rabies), and

highly unlikely in the case of most other systemic diseases. However, in the past, and given the lack of data at the time, these remote levels of risk were not acceptable to the eye banking and ophthalmic community. This has led to the expansion of CI to many different pathologies, determined by eye banking associations and applied by eye banks globally (see S1 Table).

Indeed CI expanded to disease states where no transmission has ever been described, and where there is no pathophysiological data to suggest transmission. These are neurodegenerative diseases (ND), hematological malignancies, primary tumors of the central nervous system (CNS), neoplastic meningitis and lymphangitic carcinomatosis (end-stage carcinomatous infiltration respectively of the meninges and of the lymphatic vessels, usually pulmonary) (S1 Table). Melanoma was added after evidence of disseminated malignant melanoma transmission following a kerato-limbal allograft [16].

In other disciplines, the management of chronic organ shortages has led to evidence-based reviews of donor selection and recipient allocation criteria. For example, there are age-matched donor-recipient rules for kidneys [17] and an allowance for HIV positive donor organs to be transplanted to HIV positive recipients [18] These measures have been adopted following contemporary reviews of risk (and acceptance of risk) versus the benefit derived from having the availability of the organ for transplant.

Here we report on the relative contribution of each of the CIs towards corneal donation rates following almost four years of prospective collection of data from a large University hospital in France. We then examine the validity and application of a number of these CIs and their impact on potential donation rates. Finally, we consider whether a contemporary review of decades old CI standards are warranted.

## Materials and methods

We used exhaustive data from the computerized registry of the hospital coordination for organ and tissue procurement at Saint-Etienne University Hospital (1,200 medical- surgical-obstetric beds) over a 45-month period, from 01/01/2020 to 06/09/2023, collected prospectively on a continuous basis. All deaths were recorded. For each death, the coordination team noted whether an approach of the next of kin for donation authorization had been carried out, if not, the reasons why no procedure had been carried out, any medical contraindications to donation, the type of interview with next of kin and its outcome (refusal or acceptance). The search for CI to donation was carried out on the computerized medical record and, whenever possible, by questioning the doctors who had taken care of the patient prior to his death (hospital doctor or attending physician). We have only taken into account one CI per person, since in practice, one was enough to stop the procedure.

## Results

### Analysis of contraindications

Of the 2349 deceased files analyzed, more than half (57%, n = 1346) had an CI for donation (Table 1). The identification of a ND was the most frequent, present in 16% of the files examined, representing 29% of all CI. The five diseases or families of diseases raising questions among ophthalmology experts (ND, hematological malignancies, melanomas, neoplastic meningitis and lymphangitic carcinomatosis, and CNS tumors) concerned 712 donors, corresponding to 30% of the files examined, or 53% of CI.

For ND, detailed in **Table 2**, three-quarters (75%, n = 291) were represented by cognitive disorders. It should be noted that among these, the kidneys of people who died of Huntington's chorea and amyloid angiopathy were transplanted.

**Table 1. Details of the 1346 contraindications to corneal donation identified among the 2349 files examined.**

| Contraindications | n | % |
|---|---|---|
| **Neurodegenerative diseases** | **386** | **28.7** |
| COVID 19-related | 355 | 26.3 |
| **Malignant hematological disease** | **217** | **16.1** |
| Infection | 84 | 6.2 |
| Treatment-related | 46 | 3.4 |
| Death-to-procurement time | 44 | 3.3 |
| **Melanoma** | **40** | **3.0** |
| **Neoplastic meningitis and lymphangitic carcinomatosis** | **35** | **2.6** |
| **Tumor of the nervous central system** | **34** | **2.5** |
| Serology | 33 | 2.5 |
| Ophthalmological diseases | 18 | 1.3 |
| Hemodilution | 14 | 1.0 |
| Age under 18 | 8 | 0.6 |
| Not detailed | 7 | 0.5 |
| Other transplantation | 6 | 0.4 |
| Cachexia | 5 | 0.4 |
| Deaths of unknown cause | 4 | 0.3 |
| Toxicomania | 3 | 0.2 |
| Systemic disease | 2 | 0.1 |
| Neurosurgery < 2001 | 1 | 0.1 |
| Down's syndrome | 1 | 0.1 |

In bold, the five groups of diseases we question. The categories were mutually exclusive. Each individual was counted only once.

Of the 1003 deceased without an CI for corneal donation, almost three-quarters (74%, n = 738) were the subject of an interview with their next of kin to find out whether they had any opposition. For the other 265 patients, the reasons for not meeting their relatives were mainly context-related (62%, n = 164): language barrier, spirituality, conflictual care situation or difficult relationship with the care team (**Table 3**). In 52% (n = 385) of the interviews with relatives, the non-opposition was verified and resulted in corneal procurement.

A simulation based on the lifting of CIs for the five targeted disease groups would have identified 712 more potential donors (**Fig 1**). Assuming that the rates of interview and non-

**Table 2. Distribution of different neurodegenerative pathologies considered as a contraindication to corneal donation (out of 386 cases).** The categories were mutually exclusive.

| Neurogenerative disease | n | % |
|---|---|---|
| Cognitive disorders | 291 | 75.4 |
| Not detailed | 62 | 16.1 |
| Alzheimer's disease | 9 | 2.3 |
| Creutzfeldt-Jakob disease | 9 | 2.3 |
| Parkinson's disease | 8 | 2.1 |
| Amyotrophic lateral sclerosis | 3 | 0.8 |
| Cerebral amyloid angiopathy | 2 | 0.5 |
| Multiple system atrophy | 1 | 0.3 |
| Huntington's disease | 1 | 0.3 |

**Table 3. Reasons why relatives of the deceased were not contacted.** The categories were mutually exclusive.

| Reason | n | % |
|---|---|---|
| Context | 164 | 61.9 |
| Departure of the body (by the undertaker) | 59 | 22.3 |
| Family unreachable on time | 25 | 9.4 |
| Medico-legal obstacle discovered late | 11 | 4.2 |
| Next of kin not notified of death | 4 | 1.5 |
| Thanatopraxy treatment already carried out | 2 | 0.8 |

opposition were identical to those observed over the study period, 273 additional donors could have been collected, representing a potential increase of 71%.

## Discussion

This work makes it possible to quantify the burden of CI to corneal donation at a time when the already evident global shortage of donors continues to grow. To our knowledge it is the first time such data has been published.

Each stage of the corneal donation process, from notification of a potential donor, donor selection, consent to donation, through to the retrieval of the tissue, encounters multiple obstacles (societal, legal, spiritual, logistical). These multiple constraints limit the number of tissues donated, but the impact of each varies greatly depending on the regulation, organization, and demographics of corneal donation at national, jurisdictional and local levels. While much work has been done to overcome these barriers and to improve donation rates within jurisdictions, not all approaches are amenable to implementation globally and therefore they have not had a great impact on overall global activity. However, as our data demonstrates, a reasoned reduction in the list of CIs and better-defined and better-implemented remaining CIs for donation would have an immediate and universal impact.

The cornea is unique in the panorama of transplantation. In 99.9% of cases, only its central part is transplanted. This tissue is deprived of blood and lymphatic vessels and contains no neuronal cell bodies (only severed terminal axons that degenerate after a few days storage) for full thickness keratoplasty. No neuronal tissue is transplanted at all with endothelial lamellar keratoplasty. Compared to other transplants the mass of tissue transplanted is minuscule with a full thickness graft weighing approximately 50 mg and an endothelial keratoplasty approximately 5 mg. In addition, the vast majority of recipients never receive systemic immunosuppression.

In this study, 712 of 2349 potential donor files examined (30%) had a contraindication to donation that fell within five groupings (ND, malignant hematological disease, melanoma, neoplastic meningitis and lymphangitic carcinomatosis and tumors of the CNS). These five disease groupings constituted 53% of all CIs detected. The large impact these CIs have on corneal donation, together with the validity of the medical or scientific basis of these CIs, warrants further examination and discussion on the validity on maintaining these CIs in their current form.

In the absence of any other evidence, the worldwide adoption of hemotological malignancies as CI appear to have arisen from the adoption of the Eye Bank Association of America Medical Standards first formulated in the 1980s. The original basis for excluding leukemias and lymphoproliferative orders is unclear, although it may have been based on concerns about widespread dissemination of these disorders throughout the body or the theoretical risk of a viral etiology for some of these disorders [19] However, while ocular and oculo-cerebral lymphomas and leukemic cells may be found in intraocular fluids in cases of extreme hypercytosis,

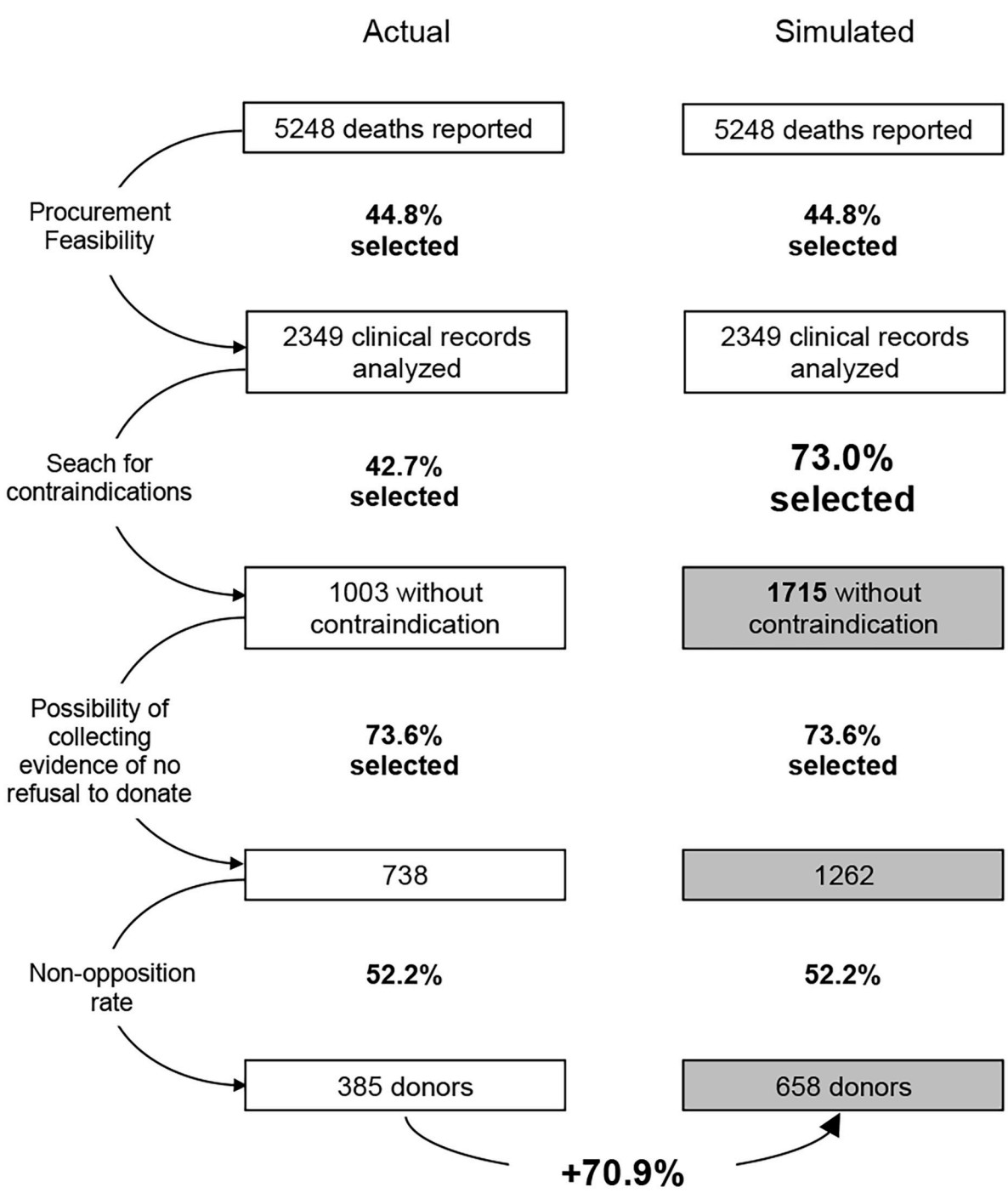

**Fig 1. Number of donors procured using the current selection procedure and the one simulated by eliminating the 5 contraindications we question.**

no corneal involvement has ever been reported in the literature, and the risk of transmission remains purely theoretical (as do cases of the disseminated diseases lymphangitis or carcinomatous meningitis). Cases of human-to-human transmission of hematological malignancies are exceptional and have only occurred in the case of vascularized organ transplants [20] The risk in corneal transplantation appears to be negligible or non-significant.

For tumors of the CNS, the precautionary principle of CI seems to be based on the embryological origin of the cornea (neural crest mesenchyme) or/and on the eye's proximity to the CNS. Extra- cranial metastases of CNS tumors are exceptional [21] and no corneal or ocular involvement has ever been reported. In comparison, metastatic adenocarcinoma has been described in the eye, but adenocarcinoma itself is not a CI for corneal donation. The transmission of a CNS tumour by transplantation remains the exception that has only been described recently for vascularised organs [22] In contrast, cases of absence of transmission from donors who died from tumors of the CNS have also been reported [23] Overall, the risk of CNS tumors transmission through corneal transplantation appears therefore to be no greater than for other neoplasms which are not CI to donation.

Melanoma was added to the list of CIs following the transmission of a disseminated malignant melanoma in a vascularized kerato-limbal allograft - a situation that requires general immunosuppression [16] Interestingly, a breast cancer transmitted under the same circumstances did not receive the same "sanction" [24] Of particular note, ocular metastases from melanoma are rare (2% in a 2023 report [25]) and with the exception of one case of corneal involvement by extension of conjunctival melanoma [26] melanoma metastases have never been reported in the cornea. In addition, a series of transplants with donor tissue from eyes with primary choroidal melanoma did not result in any transmission [27] and two corneas transplanted from a multimetastatic donor did not transmit the donor's melanoma [28] The risk for corneal transplantation appears to be non-significant. Removal as a CI would also preclude the need for inefficient and inherently inaccurate (by any but a specialist dermatologist) post-mortem skin examination to detect suspect skin lesions.

The extension of the CI for CJD to encompass all ND was based on both a concern that many of these diseases may be associated with transmissible prion disease and that a potential differential diagnosis for many of these diseases was CJD (thus risking missing contraindicating a true CJD). However, over the past decades the characterization, differentiation and diagnosis of ND has improved remarkably but were never taken into account to revise the list of contraindications. While in the past there has been some debate for and against human-to-human transmissibility [29, 30], the Institute of Medicine's (IOM's) Forum on Neuroscience and Nervous System Disorders concluded in 2013 that "There is no evidence as yet that pathogenic proteins in neurodegenerative diseases are infectious and thus spread between individuals" [31] Similarly, reservations originally expressed about the use of organs from deceased donors of amyotrophic lateral sclerosis (ALS) in the presence of immunosuppression in the recipient were dismissed upon review in 2012, with the conclusion, "no evidence exists for person- to-person transmission of neurodegenerative disease, and it is extremely unlikely that normal environmental exposures to disease-afflicted individuals could result in acquired disease." [32] While experimentally and through neurosurgical procedures there is rare evidence of transmission of some ND through direct transfer of CNS derived tissue, there is no evidence of transmission by transplantation and no evidence of transmission by any ophthalmic procedure. The risk of ND in general appears to be insignificantly low.

Notably, three-quarters of the CIs for ND in this study concerned cognitive disorders, in response to the designation "Persons with a history of rapidly progressive dementia: continuous cognitive impairment for less than 2 years" (S1 Table). This criteria (and timeline of two years) appears to derive from a limited interpretation of the WHO diagnostic criteria for CJD

in 1998 (and excludes the additional necessary criteria of myoclonus, pyramidal or extrapyramidal tract signs, cerebellar symptoms, and an akinetic mute state [33]) The CI for ND requires updating and review (and be supported by guidelines) to be more specific in its intent of excluding possible prion disease rather than its current definition which allows for the unnecessary exclusion of conditions such as Alzheimer disease or senile dementia. We suggest the 2017 criteria of "Rapidly progressive cognitive impairment with at least two of myoclonus, visual or cerebellar problems, pyramidal or extrapyramidal features, or akinetic mutism of duration less than 2 years [34].

Thus, iatrogenic transmission of disease by corneal transplantation is extraordinarily rare (with more than half of all possible cases (8/13) reported being rabies). The risk presented by the five groupings discussed, as they stand, remains a theoretical risk that has never been realized. Actual risk of transmission cannot be calculated because there have been no cases of transmission. Yet analysis and modelling of our data indicates a possible 70.9% increase in benefit if these CIs were in some instances refined, and in other instances abandoned all together. An additional 273 donors (potentially 546 corneas) could have been realized over a four-year period from a single hospital donation program. By way of comparison, the risk of Stevens-Johnson syndrome/toxic epidermal necrolysis, potentially fatal, potentially blinding, during treatment with allopurinol or cotrimoxazole is estimated at 3 per 100,000 new users [35] Yet there is no consideration of the withdrawal of these agents for therapeutic purposes; the risk is considered acceptable because of the realized benefits.

In our opinion, the excessive number of medical contraindications to cornea donation has at least 2 other harmful consequences: 1/ it needlessly overloads the work of the coordination teams. A reasoned reduction of this list would enable them to focus on the search for truly dangerous diseases. In this respect, our study clearly demonstrates that signs of CJD are drowned out by the "cognitive decline" item; 2/ it unnecessarily stigmatizes these donors and their relatives, who are denied the opportunity to generously donate.

Of the corneas distributed by the Saint-Etienne Cornea Bank between 2018–2022 (1,449 corneas to 17 transplant centers across France) just over a third went to recipients aged 75 and over. In France, life expectancy at age 75 is 12 years for men and 13 years for women (French National Institute for Statistics and Economic Studies 2022 data). At this age, there is therefore a certain urgency to receive a corneal transplant that could reduce disability, dependance and improve the quality of life. We therefore propose to carry out a large- scale, multicenter clinical trial in which corneas from donors that would have been previously contraindicated by one of the 5 CI groupings, be allocated to recipients aged 75 and over, without systemic immunosuppressive treatment. Annual follow-up will be carried out to determine if any of these diseases have developed in the recipient. The considerable increase in the pool of donors will enable these recipients to benefit from a transplant without delay and have the additional benefit of providing greater availability of corneas for other patients waiting for transplantation. All evidence to date suggests that this presents no significant increase in risk to the recipients while greatly increasing benefit.

## Conclusion

Our data shows that the potential donor pool of corneal donors is significantly restricted by a group of CIs that were introduced decades ago in response to a theoretical risk of transmission of disease. In the ensuing years this risk has not been realized. Instead a substantive amount of evidence (epidemiological, pathological, vectors of transmission) now suggests that many CIs regarding corneal donation and allocation need to be reviewed and modified or discarded altogether.

Such an approach will result in a highly significant increase in benefit with no significant increase in risk. We have shown that the number of corneas available for transplant worldwide would increase considerably and have a major impact in reducing corneal blindness. No other action, whether it be improved donor programs or cell and tissue therapy, is likely to have such a significant or rapid impact.

## Supporting information

**S1 Table. Comprehensive list of contraindications to corneal donation, classified by major category.** These recommendations comply with the selection criteria applicable to tissue donors as set out in Annex I of European Directive EC 2006/17 and Annex II of the Order of November 4, 2014. In red are those we question.
(DOCX)

**S1 Data.**
(XLSX)

## Author Contributions

**Conceptualization:** Magali Epinat, Fanny Collange, Jean Luc Perrot, Graeme Pollock, Gilles Thuret.

**Data curation:** Oliver Dorado-Cortez, Sylvain Poinard, Magali Epinat, Sandrine Ninotta, Gilles Thuret.

**Formal analysis:** Oliver Dorado-Cortez, Magali Epinat, Jean Luc Perrot, Graeme Pollock, Gilles Thuret.

**Investigation:** Oliver Dorado-Cortez, Sandrine Ninotta.

**Methodology:** Gilles Thuret.

**Supervision:** Philippe Gain.

**Validation:** Paul Goin, Graeme Pollock, Gilles Thuret.

**Visualization:** Fanny Collange.

**Writing – original draft:** Oliver Dorado-Cortez, Sylvain Poinard, Magali Epinat, Fanny Collange, Sandrine Ninotta, Paul Goin, Graeme Pollock, Gilles Thuret.

**Writing – review & editing:** Fanny Collange, Jean Luc Perrot, Philippe Gain, Graeme Pollock, Gilles Thuret.

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
