## [Decision Letter · Decision Letter 0]

9 Sep 2024

PGPH-D-24-01468

The burden of medical contraindications to corneal donation: time for review

Dear Dr. Thuret,

Thank you for submitting your manuscript to PLOS Global Public Health. After careful consideration, we feel that it has merit but does not fully meet PLOS Global Public Health’s publication criteria as it currently stands. Therefore, we invite you to submit a revised version of the manuscript that addresses the points raised during the review process.

Please note that we have only been able to secure a single reviewer to assess your manuscript. We are issuing a decision on your manuscript at this point to prevent further delays in the evaluation of your manuscript. Please be aware that the editor who handles your revised manuscript might find it necessary to invite additional reviewers to assess this work once the revised manuscript is submitted. However, we will aim to proceed on the basis of this single review if possible.

The reviewer has commented on several aspects of your manuscript, and has provided some suggestions on how these can be addressed. Please ensure you address each of the reviewer's comments when revising your manuscript.

We look forward to receiving your revised manuscript.

Kind regards,

Hugh Cowley

Staff Editor

Journal Requirements:

Additional Editor Comments (if provided):

Reviewers' comments:

Reviewer's Responses to Questions

**Comments to the Author**

1. Does this manuscript meet PLOS Global Public Health’s publication criteria? Is the manuscript technically sound, and do the data support the conclusions? The manuscript must describe methodologically and ethically rigorous research with conclusions that are appropriately drawn based on the data presented.

Reviewer #1: Yes

2. Has the statistical analysis been performed appropriately and rigorously?

Reviewer #1: Yes

3. Have the authors made all data underlying the findings in their manuscript fully available (please refer to the Data Availability Statement at the start of the manuscript PDF file)?

Reviewer #1: Yes

4. Is the manuscript presented in an intelligible fashion and written in standard English?

Reviewer #1: Yes

5. Review Comments to the Author

Reviewer #1: The authors present a compelling argument that there is sufficient evidence to revisit regulatory restrictions on eye donation that were instituted long ago. These restrictions may be creating artificial barriers to treatment and cornea accessibility with little to no increase in recipient safety. This paper raises important questions for the eye banking and corneal transplant community. I believe, with minor revisions, this paper should be published.

Line 155. This line is a little awkward. Is the procedure “a view to donation” an approach of the next of kin for donation authorization?

Carcinomatous meningitis and lymphangitis are not commonly used terms in the US. I presume others might wish to have these diseases defined so that a broader audience can understand the CIs described a little better.

DISCUSSION

Your results might be different without COVID. That said, the impact on the additional donors potentially suitable for transplant would not change. For example, 53% of the CI are part of the five Diseases (5D) you posit should be reconsidered. Without COVID as a significant cause of death in your donor pool, 5D would account for a higher percentage of ruled out donors.

Line 235. Eye Banking of America should be “Eye Bank Association of America”.

Line 274-304. Arguments for differentiating between CJD and Alzheimer disease must rely on observable clinical manifestations of the diseases. You do address this later in the section, but I think you can state this more succinctly. I recommend removing some of the discussion on the basic science related to the brain pathology.

Line 286. Please ensure that there is a space between that and normal.

Line 305. You discuss melanoma transmission as related to KLAL, but not breast CA which was reported in this paper: https://pubmed.ncbi.nlm.nih.gov/28476053/

Line 313-317. I feel like this is an odd comparison. With Allopurinol, there may not be other alternative therapies. With cornea transplant, we could do a number of things to increase tissue availability without adjusting the CIs. For example, in the US, we discard many thousands of corneas due to hepatitis B core antibody testing which is required by the FDA. The risk from these corneas is negligible. But tell that to the FDA when they know we have access to more corneas available in the US than needed in the US. Negligible risks become unacceptable risks. This could be factored into the rationale of the French mindset. Risk assessment is likely based on local need and not on global demand.

Line 327. Add “to” before “determine”

One item I didn’t see mentioned in the discussion is the inefficiency and waste incurred by the system when CIs are introduced. Our staff spends a lot of time researching disease states to ensure that we don’t inadvertently transplant tissue that is unsuitable based on our regulations. The fewer diseases to worry about, the easier it is to ensure that the diseases that really do pose a risk to recipients are not present. Unfortunately, we find things in the record after the fact that rule recovered tissue ineligible and this adds a lot of burden to the health care system and it is unfortunate when this happens for the donor and the family that gave permission to proceed. Additionally, people who want to donate tissues as an altruistic act may be excluded by these rules – we should not be creating artificial barriers to do good in the world. Only address these items in your discussion if you feel they would add to your paper.

SUPPLEMENT

Is listing herpes really absurd if it is noted in the corneal history? I would exclude that tissue. You are correct that it would be rare to have this history (or access to it), but it is not as rare as some of the other diseases noted. This comment should probably be removed as it doesn’t really advance your thesis.

6. PLOS authors have the option to publish the peer review history of their article (what does this mean?). If published, this will include your full peer review and any attached files.

**Do you want your identity to be public for this peer review?** For information about this choice, including consent withdrawal, please see our Privacy Policy.

Reviewer #1: No

---

## [Decision Letter · Decision Letter 1]

9 Oct 2024

PGPH-D-24-01468R1

The burden of medical contraindications to corneal donation: time for review

Dear Dr. Thuret,

Thank you for submitting your manuscript to PLOS Global Public Health. After careful consideration, we feel that it has merit but does not fully meet PLOS Global Public Health’s publication criteria as it currently stands. Therefore, we invite you to submit a revised version of the manuscript that addresses the points raised during the review process.

Although Reviewer 1 indicated they were satisfied with your revision, I was newly assigned your paper as an academic editor and asked to review your paper in light of the fact that only one external peer reviewer had been identified. I agree that paper is suitable for publication in PLOS Global Public Health, but believe the manuscript can be strengthened by addressing my comments below.

We look forward to receiving your revised manuscript.

Kind regards,

W. Alton Russell, PhD

Academic Editor

Journal Requirements:

Additional Editor Comments (if provided):

1. Please clarify your approach to classifying decedents with >1 contraindication. Did you capture all contraindications for each decedent, or only one? If only one was recorded, how was it chosen? If some of the patients who were contraindicated due to one of the conditions in your five groups had another contraindication, couldn't your estimate that removing these groups would increase the supply by 71% could be slightly overestimated?

2. For each table reporting n (%) please clarify in the table caption whether categories are mutually exclusive or if some individuals are counted in more than one category.

3. Supplemental Figure 1 is really a table, so should be called supplemental table 1

4. A space is missing between words on lines 145 and 200

5. (optional) You might consider addressing reviewer 1’s comment on estimating the relative contribution of your five ‘disease groups in question’ quantitatively. You can quantify the proportion of Cis that would have been due to one of your five diseases groups without COVID, and you could estimate what the percent increase in donated corneas would have been if it weren’t for COVID.

Reviewers' comments:

Reviewer's Responses to Questions

**Comments to the Author**

1. If the authors have adequately addressed your comments raised in a previous round of review and you feel that this manuscript is now acceptable for publication, you may indicate that here to bypass the “Comments to the Author” section, enter your conflict of interest statement in the “Confidential to Editor” section, and submit your "Accept" recommendation.

Reviewer #1: All comments have been addressed

2. Does this manuscript meet PLOS Global Public Health’s publication criteria? Is the manuscript technically sound, and do the data support the conclusions? The manuscript must describe methodologically and ethically rigorous research with conclusions that are appropriately drawn based on the data presented.

Reviewer #1: Yes

3. Has the statistical analysis been performed appropriately and rigorously?

Reviewer #1: Yes

4. Have the authors made all data underlying the findings in their manuscript fully available (please refer to the Data Availability Statement at the start of the manuscript PDF file)?

Reviewer #1: Yes

5. Is the manuscript presented in an intelligible fashion and written in standard English?

Reviewer #1: Yes

6. Review Comments to the Author

Reviewer #1: Thank you for addressing the concerns raised in the original review. I hope your article will prompt discussion and dialoge in the transplant community that may ultimately lead to reconsidering regulatory rules that add little or no benefit to the safety of our donor pool.

7. PLOS authors have the option to publish the peer review history of their article (what does this mean?). If published, this will include your full peer review and any attached files.

**Do you want your identity to be public for this peer review?** For information about this choice, including consent withdrawal, please see our Privacy Policy.

Reviewer #1: No

---

## [Editor Report · Decision Letter 2]

5 Nov 2024

The burden of medical contraindications to corneal donation: time for review

PGPH-D-24-01468R2

Dear Pr. Thuret,

We are pleased to inform you that your manuscript 'The burden of medical contraindications to corneal donation: time for review' has been provisionally accepted for publication in PLOS Global Public Health.

Best regards,

W. Alton Russell, PhD

Academic Editor